# Detection of *EGFR* Mutations in Plasma Cell-Free Tumor DNA of TKI-Treated Advanced-NSCLC Patients by Three Methodologies: Scorpion-ARMS, PNAClamp, and Digital PCR

**DOI:** 10.3390/diagnostics10121062

**Published:** 2020-12-07

**Authors:** Annamaria Siggillino, Paola Ulivi, Luigi Pasini, Maria Sole Reda, Elisa Chiadini, Francesca Romana Tofanetti, Sara Baglivo, Giulio Metro, Lucio Crinó, Angelo Delmonte, Vincenzo Minotti, Fausto Roila, Vienna Ludovini

**Affiliations:** 1Medical Oncology Division, S. Maria della Misericordia Hospital, 06132 Perugia, Italy; annamaria.siggillino@ospedale.perugia.it (A.S.); mariasole.reda@ospedale.perugia.it (M.S.R.); francesca.tofanetti@ospedale.perugia.it (F.R.T.); sara.baglivo@ospedale.perugia.it (S.B.); giulio.metro@ospedale.perugia.it (G.M.); vincenzo.minotti@ospedale.perugia.it (V.M.); fausto.roila@ospedale.perugia.it (F.R.); vienna.ludovini@ospedale.perugia.it (V.L.); 2Biosciences Laboratory, Istituto Scientifico Romagnolo per lo Studio e la Cura dei Tumori (IRST) IRCCS, 47014 Meldola, Italy; paola.ulivi@irst.emr.it (P.U.); elisa.chiadini@irst.emr.it (E.C.); 3Department of Medical Oncology, Istituto Scientifico Romagnolo per lo Studio e la Cura dei Tumori (IRST) IRCCS, 47014 Meldola, Italy; lucio.crino@irst.emr.it (L.C.); angelo.delmonte@irst.emr.it (A.D.)

**Keywords:** *EGFR*, TKIs, cftDNA, liquid biopsy, NSCLC

## Abstract

Analysis of circulating cell-free tumor DNA (cftDNA) has emerged as a specific and sensitive blood-based approach to detect epidermal growth factor receptor (*EGFR*) mutations in non-small cell lung cancer (NSCLC) patients. Still, there is some debate on what should be the preferential clinical method for plasma-derived cftDNA analysis. We tested 31 NSCLC patients treated with anti-EGFR tyrosine kinase inhibitors (TKIs), at baseline and serially during therapy, by comparing three methodologies in detecting *EGFR* mutations (L858R, exon 19 deletion, and T790M) from plasma: scorpions-amplification refractory mutation system (ARMS) methodology by using *EGFR* Plasma RGQ PCR Kit-QIAGEN, peptide nucleic acid (PNA) clamp and PANA RealTyper integration by using PNAClamp *EGFR*-PANAGENE, and digital real time PCR by using QuantStudio 3D Digital PCR System-Thermo Fisher Scientific. Specificity was 100% for all three mutations, independently from the platform used. The sensitivity for L858R (42.86%) and T790M (100%) did not change based on the method, while the sensitivity for Del 19 differed markedly (Scorpion-ARMS 45%, PNAClamp 75%, and Digital PCR 85%). The detection rate was also higher (94.23%) as measured by Digital PCR, and when we monitored the evolution of *EGFR* mutations over time, it evidenced the extreme inter-patient heterogeneity in terms of levels of circulating mutated copies. In our study, Digital PCR showed the best correlation with tissue biopsy and the highest sensitivity to attain the potential clinical utility of monitoring plasma levels of *EGFR* mutations.

## 1. Introduction

The identification of activating epidermal growth factor receptor (*EGFR*) mutations plays an important role in determining the treatment response to EGFR tyrosine kinase inhibitors (TKIs), including gefitinib, erlotinib, afatinib and, more recently, osimertinib in advanced non-small cell lung cancer (NSCLC) patients. The most common activating mutations of the *EGFR* gene are the in-frame deletions of exon 19 and the missense mutations of exon 21 (i.e., p.Leu858Arg), constituting more than 90% of known *EGFR* activating mutations [1,2]. Unfortunately, almost all patients with NSCLC who respond to EGFR-TKIs therapy soon develop acquired resistance and experience disease progression within 10 to 16 months. The T790M mutation in *EGFR* exon 20 is a recurrent mechanism of resistance to first-line EGFR-TKIs, detectable in nearly 50% of tissue specimens at progression [3,4,5]. The frequency of T790M mutation in EGFR-TKI-naive patients and its dynamic changes during therapy remains unclear [6,7,8]. The third-generation EGFR-TKI osimertinib, which specifically targets *EGFR* T790M mutation, was approved for use in some countries, including the United of States (US) and the European Union (EU), in patients who have developed T790M after first and second generation TKI treatment and, more recently, in EGFR mutated treatment naïve patients. Hence, it is of increasing importance to collect information from serial biopsies on T790M to determine the appropriateness of osimertinib treatment [9,10]. However, repeated tissue biopsies in patients with advanced disease is not always feasible, due to the invasiveness of the intervention, and when it is possible it may be difficult to obtain enough tumor DNA for the *EGFR* mutation test. Moreover, tissue specimens may not be reflective of the patient’s complete disease burden due to spatial and temporal tumor heterogeneity [11,12]. In recent years, new methods to detect disease relevant mutations from liquid biopsy as alternative sources are being developed. In particular, circulating cell-free tumor DNA (cftDNA) in plasma has emerged as a specific and sensitive blood-based biomarker for the detection of *EGFR* mutations. Genotyping cftDNA in a fresh blood sample represents a noninvasive and feasible method for real-time monitoring of the treatment response to EGFR-TKIs and to predict drug resistance [13,14,15,16,17,18,19]. Moreover, *EGFR*-activating mutation analysis on cftDNA has been approved as a companion diagnosis to select NSCLC patients for treatment with gefitinib and osimertinib in the EU. However, some technical limitations in detecting *EGFR* mutations with cftDNA have been reported. For example, the quantity and quality of circulating tumor-derived DNA varies widely between patients [20]. Specifically, the abundance of cftDNA varies from 0.01% to 67% for patients with different kinds of cancers or progression stages [21,22]. Several studies have also evaluated the concordance between mutations detected in tumor tissues and those observed in plasma cftDNA with different sensitivity results depending on the type of technology used [14,23,24,25,26]. DNA from normal cells is always present in plasma together with tumor-released cell free DNA, which often represents only a small fraction of the total circulating DNA, so it is important to use high sensitive technologies to detect tumor-specific somatic mutations. Among several methodologies, such as amplification refractory mutation system (ARMS), Digital PCR and next-generation sequencing (NGS) [27,28,29], there is not a widely accepted and approved method for *EGFR* mutation analysis from cftDNA. In this study, we reported a performance comparison between three technologies (Scorpion-ARMS EGFR Plasma RGQ PCR Kit-QIAGEN, QuantStudio 3D Digital PCR System-Thermo Fisher Scientific and PNAClamp EGFR-PANAGENE) in detecting clinically-relevant *EGFR* mutations in tumor tissue and plasma collected from NSCLC patients. We monitored *EGFR* mutations in plasma samples at baseline and serially during treatment with EGFR-TKIs to predict early development of resistance to treatment.

## 2. Materials and Methods

### 2.1. Patient Selection

This was a prospective, multi-institution clinical study that included NSCLC patients treated at the Medical Oncology Division in S. Maria della Misericordia Hospital of Perugia, and at Istituto Scientifico Romagnolo per lo Studio e la Cura dei Tumori (IRST) IRCCS of Meldola (FC), Italy, from November 2014 to July 2017. Patients were considered for inclusion if they met the following criteria: (1) pathologically confirmed diagnoses with advanced or recurrent primary lung cancer according to the seventh Edition of TNM in Lung Cancer [30], with sufficient tissue samples for the study harboring activating EGFR mutations and sufficient peripheral blood; (2) the possibility to make serial blood samples during the TKIs treatment (baseline, 8 days and 20 days after the start of treatment, clinical evaluation and progression of the disease, if occurred during the sampling); (3) complete information obtained, including age, gender, smoking history, and staging; (4) complete medical documentation including follow-up records followed by an eventual systemic objective progression according to response evaluation criteria in solid tumors or World Health Organization criteria (RECIST). Exclusion criteria included: (1) aged below 18 years old; (2) pregnant patients. The study was conducted in accordance with Declaration of Helsinki principles, and approved by the Institutional Review Board of the S. Maria della Misericordia Hospital of Perugia, (Number: 2576/15, approved April 28, 2015 and IRST of Meldola (FC) (Number: 1297/15; approved March 18, 2015. Written informed consent was obtained from each patient prior to study entry.

### 2.2. Tissue and Blood Sample Collection and cftDNA Extraction

Tissue and blood samples were obtained from advanced NSCLC patients treated with EGFR-TKIs. Tumor tissue genomic DNA was extracted from 10 formalin fixed paraffin embedded (FFPE) slides using the QIAamp DNA FFPE Tissue Kit on the QIAcube Instrument (Qiagen S.p.A. Milan, Italy). Peripheral blood was collected into 2 tubes containing EDTA-K2 anticoagulant (5 mL) (BD Diagnostics, Buccinasco-Milan, Italy) and processed within 30 min. Whole blood was first centrifuged at 1100 g for 15 min to separate the plasma from the peripheral blood cells. The supernatant was collected and transferred into a 2 mL Eppendorf tube (EP tube), followed by centrifugation at 1500 g for 10 min to pellet any remaining cells. Plasma (supernatant) was collected, transferred into a new 2 mL EP tube and stored at −80 °C. CftDNA were extracted from the plasma samples (at least 2 mL) using QIAamp Circulating Nucleic Acid Kit on the QIAvac instrument 24 Plus (Qiagen S.p.A. Milan, Italy), according to the manufacturer’s instruction. The cftDNA was recovered in 55 μL elution buffers (TE Buffer) and immediately stored at −20 °C until use.

### 2.3. Detection of EGFR Mutations in Tumor Tissue

EGFR mutation testing on tissue samples of patients was performed at the diagnosis using two standardized methodologies, Scorpion-ARMS PCR Kit on Rotor-Gene Q MDx instrument, (Qiagen, S.p.A. Milan, Italy) or MYRIAPOD^®^ Lung status (Diatech Pharmacogenetics, Jesi, Italy) using the MASSArray Sequenom system (Diatech Pharmacogenetics). Assays were performed according to each manufacturer’s protocols.

### 2.4. Detection of EGFR Mutations in Plasma

Extracted cftDNA of patients harboring activating mutations (exon 19 deletions, L858R and T790M) in tumor tissue, were tested for the same *EGFR* mutations using three different technologies: Scorpion-ARMS *EGFR* Plasma RGQ PCR Kit-QIAGEN, PNAClamp R *EGFR*-PANAGENE and QuantStudio 3D Digital PCR System-Thermo Fisher Scientific. Assays were performed according to each manufacturer’s protocols.

### 2.5. Scorpion-ARMS EGFR Plasma RGQ

The Scorpion-ARMS EGFR Plasma RGQ PCR Kit is an in vitro diagnostic test for the detection of the 21 EGFR mutations (Del 19, L858R and T790M) on cftDNA extracted from plasma using a real-time polymerase chain reaction (Q-PCR) on the Rotor-Gene Q MDx instrument (Qiagen S.p.A. Milan, Italy). The kit utilizes two technologies, amplification refractory mutation system (ARMS) that ensures distinguishing between a match and a mismatch at the 3’ end of a PCR primer, combined with Scorpions, bifunctional molecules containing a PCR primer covalently linked to a probe to cause increased fluorescence from the reaction tube. The assays were carried out according to manufacturer’s instructions. Each run contains a positive and a negative control and each sample is analyzed for the mutations and for a control assay that amplifies a region of exon 2 of the *EGFR* gene and is used as a reference to calculate the ∆Ct. The assay provides a qualitative assessment of the mutation status.

### 2.6. PNAClamp EGFR

The PNAClamp *EGFR* kit (PANAGENE, Daejeon, Korea), a technology based on peptide nucleic acid (PNA)-mediated real time PCR clamping and melting peak analysis, was used for mutation analysis of the cftDNA. This technology integrates PNAClamp™ and PANA RealTyper™ (PNA probe-based fluorescence melting curve analysis). PNAClamp™ is a kind of PCR technology and uses peptide nucleic acid (PNA) probes which complementarily bind to the wild-type DNA. PANA RealTyper™ uses multiplex melting curve analysis with fluorescence labelled PNA probes. PNAClamp™ takes advantage of both technologies. It is not only able to detect small amounts of mutation with high sensitivity, but it is also able to genotype multiple mutations, simultaneously analyzing their own melting temperature (Tm) value for the sequence changes of the target gene.

### 2.7. QuantStudio 3D Digital PCR

Digital PCR (dPCR) was performed using the QuantStudio 3D Digital PCR platform (Thermo Fisher Scientific, Monza, Italy). Mutations analysis of cftDNA was performed by an allele specific TaqMan^®^ probe targeting known *EGFR* mutations (exon 19 deletions, L858R and T790M mutation). Wild Type (WT) EGFR alleles were represented by a VIC fluorescent probe while mutant EGFR alleles were represented by a FAM fluorescent probe. We purchased all 3D Digital PCR reagents from Thermo Fisher Scientific, the custom ordered primers were: T790M (Assay ID: AHRSROS), L858R (Assay ID:AHRSRSV) and Del 19 (Assay ID:Hs00000228_mu). The final 15 μL of TaqMan PCR reaction mixture was made up according to the following: 7.5 μL 2× QuantStudio™ 3D Digital PCR Master Mix, 0.75 μL 20× TaqManAssay (primer/probe mix), 6.75 μL diluted DNA (25 ng), and then loaded into the QuantStudio™ 3D Digital PCR Chip, which has 20,000 mini-chambers. To perform the PCR using the ProFlex™ 2× Flat PCR System, the thermal cycling profile was 10 min of incubation at 96 °C, followed by 39 cycles of 60 °C for 2 min, 98 °C for 30 s, 60 °C for 2 min, and then 4 °C hold. We used the QuantStudio™ 3D Digital PCR instrument to read the chip. The subsequent analysis was performed with the QuantStudio 3D Analysis Suite Software. The QuantStudio™ 3D Digital PCR System provides qualitative detection of target nucleic acid sequences (targets) and relative (% target/total) or absolute quantification (copies/µl) using allele specific post-PCR (endpoint) analysis.

### 2.8. Statistical Analysis

The limit of detection (LOD) of each assay was defined as the lowest target concentration that could be specifically detected and was determined for each of the three technologies using the Horizon Multiplex I cfDNA Reference Standard Set (HD780) (Horizon Discovery, Cambridge, UK). The reference standard DNAs that were used included Del 19, L858R, and T790M mutations. Each reference mutant DNA contained the mutant sequence at a frequency of 5%, 1%, and 0.1%, respectively. Reference mutant DNA was also diluted with the corresponding WT *EGFR* reference DNA to obtain 0.5% and 0.01% allele frequencies DNA. Overall survival (OS) and progression free survival (PFS) were modeled via the Cox proportional hazard regression model. Baseline continuous T790M levels were separately considered as covariates. Results are summarized as hazard ratio (HR) with their 95% confidence intervals and the corresponding p-value. Statistical analyses were achieved using the statistical language R version 3.6.3, and its packages Survminer Version 0.4.8 and survival version 3.2.3.

## 3. Results

### 3.1. Patient Characteristics

A total of thirty-one NSCLC patients were enrolled, the median age was 68 years old (range, 38–88 years). Patient characteristics at baseline are shown in Table 1. Most patients were female (74.2%), 64.5% were never-smoker, 90.4% had adenocarcinoma and 93.6% were diagnosed at stage IV. According to the results of the tumor tissue *EGFR* analysis, twenty patients (64.5%) had Del 19 mutation, six patients (19.4%) had L858R mutation, one (3.2%) had both L858R and T790M mutations and four (12.9%) had other mutations (G719S+L833V, L861Q, L858M, G719C, and S768I). All patients received an EGFR-TKI (35.5% gefitinib, 12.9% erlotinib and 51.6% afatinib) as a first-line therapy and 26 (83.8%) developed disease progression and 21 of them died.

### 3.2. Sensibility, Specificity and Coincidence Rate of the Three Methods for the Three EGFR Mutations

EGFR mutation analysis of cftDNA extracted from baseline plasma was performed with the three methodologies (Scorpion-ARMS *EGFR* Plasma, PNAClamp R *EGFR* and QuantStudio 3D Digital PCR) and compared with the *EGFR* mutation analysis of the corresponding tumor tissue sample. We measured sensitivity, and the coincidence rate of each of these three cftDNA assays, by using the tissue *EGFR* mutational profiles for the three driver mutations (Del 19, L858R, T790M) as a reference. Specificity was calculated using DNA extracted from the plasma of 12 healthy blood donors. The results showed that specificity was 100% (12/12) for all the three mutations measured by each of the three methods (Figure 1). Digital PCR showed the highest sensitivity in detecting the Del 19, compared to the other methods. The coincidence rate for Del 19 was also different, based on the method used. Sensitivity for Del 19: Scorpion-ARMS 45% (9/20), Digital PCR 85% (17/20), and PNAClamp 75% (15/20); coincidence rate for Del 19: Scorpion-ARMS 65.62% (21/32), Digital PCR 90.62% (29/32), and PNAClamp 84.37% (27/32). For both L858R and T790M sensitivity and coincidence rates were the same no matter what type of method was used. Sensitivity for L858R was 42.86% (3/7) and 100% (1/1) for T790M. The coincidence rate for L858R was 78.95% (15/19) and 100% (13/13) for T790M.

### 3.3. Comparison of the Three Methodologies in Detecting EGFR Mutations in Plasma versus Tissue

Figure 2 represents the results (Appendix A) for *EGFR* mutational analysis in tissue biopsies compared to plasma samples, along with the type of TKI administration and relative best response for each patient (*n* = 31). The three platforms in liquid biopsy detected at a baseline of a total of three patients with the L858R mutation, seventeen with the Del 19 mutations, and seven with the T790M mutations, compared to seven patients with L858R, twenty with Del 19, and one patient with T790M mutations present in the tumor tissue. Six patients were found positive for the T790M mutation as detected in plasma by Digital PCR while testing negative in tissue analysis (Pt ID.1, 2, 5, 9, 23, 24). In four of these patients the T790M was co-present with either Del 19 (Pt ID. 2, 9) or L858R (Pt ID. 24, 28). In nine patients, the Del 19 mutation observed in tumor tissue was also detected with all three methods in liquid biopsy. None of these mutations was found in plasma by any of the three methods used in the four patients (Pt ID. 8, 19, 22, 27) for which tissue analysis resulted positive for either of the three activating mutations.

### 3.4. Limit of Detection of the Three Methods for Three Mutations

Analysis with the Horizon Multiplex I cfDNA Reference Standard Set (HD780) demonstrated that the limit of detection (LOD) with Digital PCR for the T790M, Del 19 and L858R mutations was 0.1%, while Scorpion-ARMS and PNAClamp could reveal Del 19 and L858R mutations at a minimal frequency of 0.5%, and T790M mutations at a frequency of 1%. (Appendix A)

### 3.5. EGFR Mutation Detection Rate in Plasma

We found a total of 104 *EGFR* mutations in the cftDNA of 31 NSCLC patients considering the three platforms together. The overall detection rate of Scorpion-ARMS *EGFR* Plasma, PNAClamp R *EGFR*, and QuantStudio 3D Digital PCR was 33.65% (35/104), 62.5% (65/104), and 94.23% (98/104) respectively (Figure 3A). The 104 *EGFR* mutations were divided into two subgroups (low copy number: < 1 copy/µL; high copy number: ≥ 1 copy/µL). The subdivision was based on the quantitative results obtained by Digital PCR (total 98 mutations).The low copy number group included 52 mutations while the high copy number group includes 46 mutations. We then compared the results obtained with either Scorpion-ARMS and PNAClamp for these subgroups. In the low copy/µL subgroup, the detection rate of Scorpion-ARMS was 11.53% (6/52) and 40.38% (21/52) for Scorpion-ARMS and PNAClamp respectively (Figure 3B). In the high copy/µL subgroup the detection rate of Scorpion-ARMS and PNAClamp was 63.04% (29/46) and 82.6% (38/46) respectively (Figure 3C). We then made the same type of comparison looking at the three main *EGFR* mutations individually (T790M, Del 19, and L858R). In the low copy/µL subgroup, the detection rate of Scorpion-ARMS and PNAClamp were respectively 4.54% (1/22) and 13.63% (3/22) for T790M, 13.04% (3/23) and 56.52% (13/23) for Del 19, and 28.57% (2/7) and 71.42% (5/7) for L858R (Figure 3D). In the high copy/µL subgroup, the detection rate of Scorpion-ARMS and PNAClamp was, respectively, 33.33% (4/12) and 66.66% (8/12) for T790M, 66.66% (16/24) and 83.33% (20/24) for Del 19, and 90% (9/10) and 100% (10/10) for L858R (Figure 3E).

### 3.6. Monitoring the Plasma Levels of T790M, Del 19, and L858R during Therapy

To assess the potential clinical utility of monitoring plasma levels of *EGFR* mutations (T790M, Del 19, and L858R) in the course of TKI treatment, we analyzed cftDNA at baseline and serially at 8 and 20 days post-treatment, at the first clinical evaluation, and at disease progression (Figure 4; Appendix A). For four of the 31 patients monitored (Pt no. 3, 20, 23, 31) was possible to measure only the T790M mutation. Seven patients (Pt no. 1, 5, 8, 17, 19, 22, 27) were negative for either the Del 19 or L858R activating mutations at baseline. Disease progression occurred in 18 patients, ten of which (Pt n. 2, 6, 9, 10, 11, 15, 16, 18, 24, 28) showed concomitant presence in liquid biopsy of the T790M resistance mutation with one of the two activating mutations (either the Del 19 or L858R) at progression. At progression, one patient (Pt no. 5) was positive only for T790M, and 3 patients (Pt no. 8, 14, 26) were positive only for the activating mutations. Interestingly, in Pt n.8 and n.26, theT790M was absent at baseline, appeared at some point during treatment, to disappear again at the time of progression. Finally, one patient (Pt no. 7) was negative at progression for either the T790M or the activating mutation, while another one (Pt no. 31) tested completely negative for the T790M at any time point. In any case, presence of T790M at baseline was significantly associated with a shorter PFS, (HR = 1.246 (1.051, 1.477); *p* = 0.011) but not OS, (HR = 1.042 (0.926, 1.173); *p* = 0.491).

## 4. Discussion

With the approval of liquid biopsy for molecular testing of patients with NSCLC [31], typically in those cases with insufficient tumor tissue or in cases where specimens are not obtainable, and with the advent of third-generation EGFR-TKI, concerns have been raised to establish which platform would be the best to accurately evaluate *EGFR* mutations. As re-biopsy has several limitations, plasma cftDNA has emerged as a new and promising approach for non-invasive genotyping, facilitating dynamic monitoring of gene mutation in the course of treatment. Several studies have been conducted comparing different platforms [32,33,34,35,36,37,38], but to date there is no standardized procedure or a reference method. The aim of this study was to compare three methodologies that are based on targeted approaches: the QuantStudio 3D Digital PCR, the Scorpion-ARMS *EGFR* Plasma RGQ PCR Kit, and the PNAClamp *EGFR*. We applied these three platforms to detect cftDNA *EGFR* mutations in the plasma of EGFR-TKI treated patients with advanced NSCLC, at baseline and serially during the treatment. Our analysis revealed that the best plasma-tissue correlation was reached by Digital PCR. As well, Digital PCR showed the highest sensitivity in detecting the Del 19 mutation, while there was no difference between the three platforms as for the L858R and T790M mutations, although the latter mutation was observed in only one case. Moreover, analysis with reference standards demonstrated that Digital PCR was the methodology that allowed detection of the three mutations at the lowest allelic frequency. Finally, Digital PCR reached the highest detection rate among the three platforms used, followed by the PNAClamp, in both the low copy/µL and the high copy/µL subgroup.

Many studies have been conducted to find the most accurate method for determining EGFR mutations in the cftDNA of lung cancer patients, including technologies such as ARMS [39,40], PNA-clamp [41], DHPLC [42], and NGS [29]. However, sensitivity and specificity varied significantly among different studies. Droplet Digital (ddPCR) has been reported as the technique with the highest sensitivity and specificity in detecting *EGFR* mutations in plasma: on average 81.82% and 98.44%, respectively [43].

This is one of the few studies comparing the QuantStudio 3D Digital PCR with other technologies, and demonstrating the high diagnostic accuracy of this method. These results are in accordance with those reported by Feng et al. [44], where the superiority of QuantStudio 3D Digital PCR was demonstrated, in comparison to ARMS-PCR, in detecting T790M *EGFR* mutations. There is an increasing necessity in clinical practice to characterize tumors for a panel of gene alterations, rather than to a single mutation, and this could be feasible only using NGS methodologies; digital PCR could represent a valid method for the monitoring of specific resistance mutations, such as T790M or others. Digital PCR is a highly sensitive, quick and low in cost method that could be very useful in the clinical practice for the monitoring of selected mutations. In view of the recent approval of the third generation TKI osimertinib in the first line treatment of EGFR mutated patients, other resistance mutations will emerge in future studies, and monitoring will become important during treatment, using methodologies like digital PCR.

Moreover, in view of recent results of the ADAURA study [45] demonstrating the efficacy of osimertinib in EGFR mutated resected NSCLC it will become essential to detect EGFR mutation using highly sensitive methodologies, as the releasing of cftDNA in early stage tumor could be very low, leading to the necessity to use methodologies with a very low limit of detection.

## 5. Conclusions

Our study investigated three different methodologies for detecting *EGFR* mutations in plasma samples of NSCLC patients. Results indicated that best correlation data with tissue and highest sensitivity was reached with Digital PCR. Furthermore, Digital PCR on cftDNA provides a promising and non-invasive assay to test *EGFR* mutations, also providing quantitative data. Compared to other methods Digital PCR would be a robust method for absolute quantification of samples in a longitudinal way, allowing better monitoring of the evolution of mutations over time. Moreover, our results highlight the potential application of liquid biopsy using Digital PCR as a routine assay in clinical practice for both detection and quantification of actionable mutation landscape in NSCLC patients.

## Figures and Tables

**Figure 1 diagnostics-10-01062-f001:**
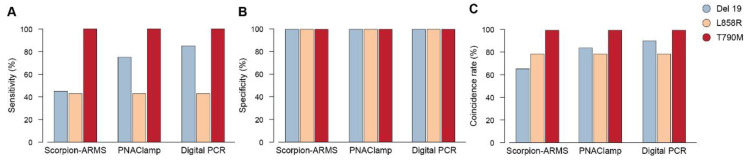
Levels of sensibility, specificity and coincidence rate of the three methods in detecting epidermal growth factor receptor (*EGFR*) mutations in plasma samples. *EGFR* mutations as detected in liquid biopsy at baseline through cell-free tumor DNA (cftDNA) analysis by the Scorpion-ARMS *EGFR* Plasma RGQ PCR Kit-QIAGEN, PNAClamp *EGFR*-PANAGENE, and the QuantStudio 3D Digital PCR System-Thermo Fisher Scientific. (**A**) Sensitivity, (**B**) specificity, and (**C**) coincidence rate of each of these three assays, were calculated on the basis of tissue analysis.

**Figure 2 diagnostics-10-01062-f002:**
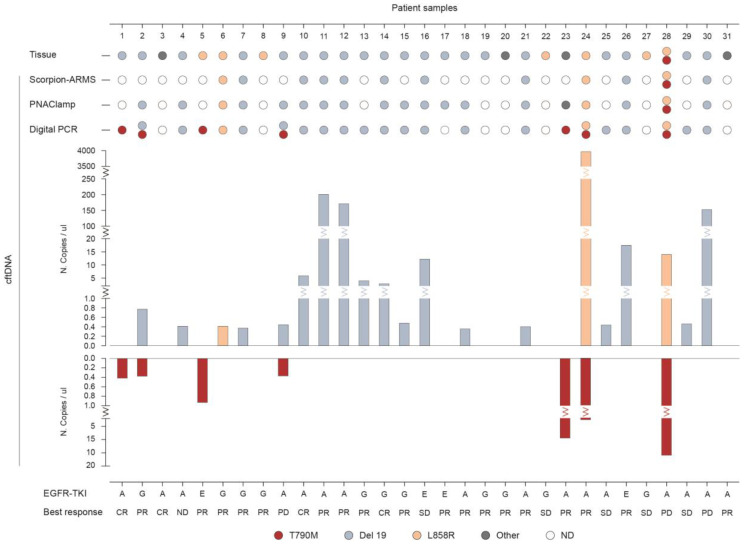
Comparison of the three methodologies in detecting *EGFR* mutations in plasma versus tissue analysis. *EGFR* mutations as detected in liquid biopsy at baseline through cell-free tumor DNA (cftDNA) analysis by Scorpion-ARMS *EGFR* Plasma RGQ PCR Kit-QIAGEN, PNAClamp *EGFR*-PANAGENE, and QuantStudio 3D Digital PCR System-Thermo Fisher Scientific. Mutational analysis of *EGFR* in plasma samples is compared to tissue biopsies (*n* = 31; ND, not determined). The number of mutated EGFR copies obtained by QuantStudio 3D Digital PCR is indicated as copies/ul. Results are presented along with the type of anti-EGFR treatment and clinical response (TKI, Tyrosin Kinase Inhibitor; A, Afatinib; G, Gefitinib; E, Erlotinib; CR, Complete Response; PR, Partial Response; SD, Stable Disease; PD, Progressive Disease).

**Figure 3 diagnostics-10-01062-f003:**
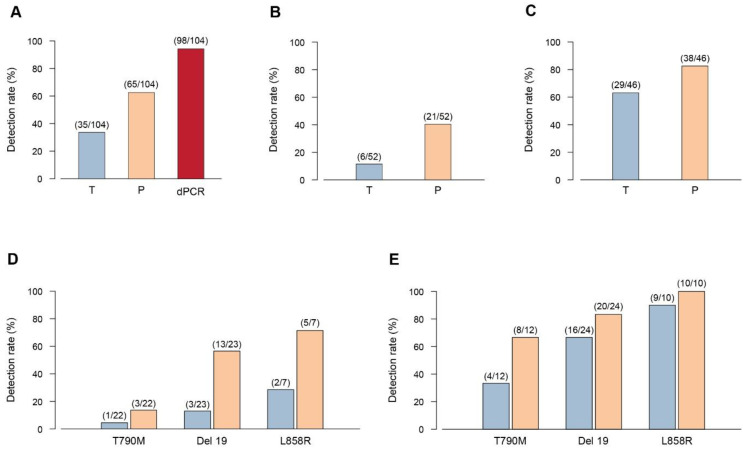
Detection rate of three platforms in plasma *EGFR* mutations detection. The detection rate is shown as the frequency of sample that resulted positive when tested in liquid biopsy with the three different methods: Scorpion-ARMS *EGFR* Plasma RGQ PCR Kit-QIAGEN (T), PNAClamp *EGFR*-PANAGENE (P), and QuantStudio 3D Digital PCR System-Thermo Fisher Scientific (dPCR). (**A**) Overall detection rate of the three platforms. (**B**) Overall detection rate of the Scorpion-ARMS compared to the PNAClamp for the low copy (<1 copy/µL) patient group as measured by Digital PCR. (**C**) Overall detection rate of Scorpion-ARMS compared to PNAClamp for the high copy (>1 copy/µL) patient group as measured by Digital PCR. (**D**) Comparison of Scorpion-ARMS (blue) with PNAClamp (orange) for the low copy (<1 copy/µL) patient group as measured by Digital PCR, considered separately for each *EGFR* main mutation (Del 19, L858R, and T790M). (**E**) Comparison of Scorpion-ARMS (blue) with PNAClamp (orange) for the high copy (>1 copy/µL) patient group, measured by Digital PCR, considered separately for each *EGFR* main mutation (T790M, Del 19, and L858R). Numbers in parenthesis indicate the relative amount of mutations detected with each specific method.

**Figure 4 diagnostics-10-01062-f004:**
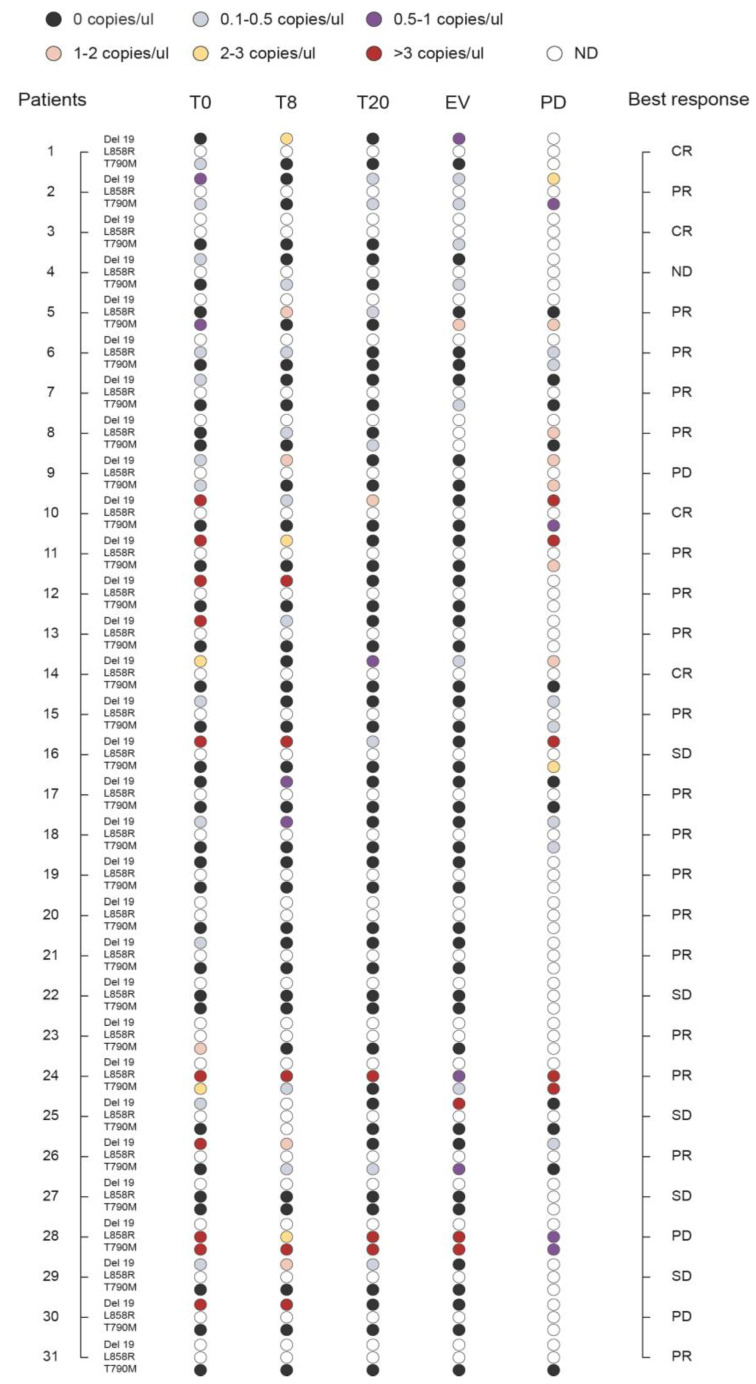
Monitoring the levels of Del 19, L858R, and T790M during therapy. Plasma levels of *EGFR* mutations (Del 19, L858R, and T790M) in the course of tyrosine kinase inhibitor (TKI) treatment were monitored by analyzing cell-free tumor DNA (cftDNA) by QuantStudio 3D Digital PCR System-Thermo Fisher Scientific at baseline (T0) and serially at 8 days (T8) and 20 days (T20) post-treatment, at the first clinical evaluation (EV), and at disease progression (PD). Quantitative measures are represented as thresholds of numbers of *EGFR* mutated copies/ul as reported in Appendix A (*n* = 31; ND, not determined). Results are presented along with the best clinical response (CR, Complete Response; PR, Partial Response; SD, Stable Disease; PD, Progressive Disease).

**Table 1 diagnostics-10-01062-t001:** Patient characteristics.

Characteristics	Patients	
	*n* = 31	%
Median Age, Years (range)	68 (38–88)	
Sex		
Female	23	74.2
Male	8	25.8
Performance Status		
0	19	61.3
1	10	32.3
2	2	6.4
Smoking History		
Never smoker	20	64.5
Former	3	9.7
Smoker	8	25.8
Histology		
Adenocarcinoma	28	90.4
Squamous-cell carcinoma	1	3.2
Adenosquamous	1	3.2
NSCLC	1	3.2
Stage		
IIIA R2	1	3.2
IIIB	1	3.2
IV	29	93.6
EGFR mutations (tissue)		
L858R	6	19.4
L858R+T790M	1	3.2
Del 19	20	64.5
Other	4	12.9
EGFR-TKI		
Gefitinib	11	35.5
Erlotinib	4	12.9
Afatinib	16	51.6
Progression		
Yes	26	83.8
No	5	16.2
Exitus		
Alive	10	32.3
Dead	21	67.7

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
