# Peer review of "Detection of EGFR Mutations in Plasma Cell-Free Tumor DNA of TKI-Treated Advanced-NSCLC Patients by Three Methodologies: Scorpion-ARMS, PNAClamp, and Digital PCR"

_diagnostics, 2020, doi:10.3390/diagnostics10121062_

Round 1
Reviewer 1 Report
Siggillino et al compared three methods (ARMS, PCR clamp, and digital PCR) to detect three EGFR mutations. Through experiments, the authors concluded that digital PCR is the most sensitive method among them. Their results would be informative for researchers and medical doctors who use those methods. However, before considering publication in Diagnostics, the authors should respond to comments described below.
1) Therascreen and PANAMutyper are product names whereas Digital PCR is a name of methods. Therefore, to avoid reader’s confusion, names of methods but not products should be used throughout the manuscript. Alternatively, if the authors insist on product names, they should describe the reason in the introduction section.
2) Where did the numbers below come from?
12/“12” in the line 96
“104” in the line 139
3) As to description in the line 133 – 137, results should be shown as Supplementary data.
4) As to Figure 2, it is difficult to interpret detection of T790M in plasma by digital PCR. How can we interpret detection of T790M in #1, #2, #5, #9, #23, and #24? Are they false positive? Dose the detection affect specificity of digital PCR?
5) As to Figure 3, it would be helpful for readers to show not only detected numbers of mutations but also examined total numbers in parentheses (e.g., 6/52, 1/22) even if examined total numbers are shown in the main text.
6) The authors should define EV and PD in Figure 4 in more detail.
7) Line 211 – 212: Only one T790M sample was tested in this study. Therefore, it would be difficult to conclude as “no difference” on detection of T790M.
8) The discussion section is like a summary of results but not discussion. The authors should discuss results and pros/cons of digital PCR in more detail in the section.
9) There is no detailed information on primer and probe sequences used for digital PCR. The information should be shown if the authors designed them by themselves. Alternatively, if the authors purchased primers and probes from Thermo Fisher Scientific as kits, those product numbers should be shown.
10) PFS, HR, and OS should be defined in the line 186–187.
11) “4” and “*” are not shown in the line 6–8.
Author Response
Answers to reviewers
Manuscript: Siggillino et al - diagnostics-1001893
We thank the reviewer for the precious suggestions and comments provided. Please find our detailed answers in the text below. Correction and chances in the main manuscript are highlighted in yellow. We have also included a modified version of Fig. 1, Fig. 2, and Fig. 3, and added the Supplementary Table S3.
REVIEWER 1
Siggillino et al compared three methods (ARMS, PCR clamp, and digital PCR) to detect three EGFR mutations. Through experiments, the authors concluded that digital PCR is the most sensitive method among them. Their results would be informative for researchers and medical doctors who use those methods. However, before considering publication in Diagnostics, the authors should respond to comments described below.
Does the introduction provide sufficient background and include all relevant references?
Reply: The introduction has been improved with some bibliographic references (number 17, 18, 19, 25,26 – lines 416-425, 437-442)
Are the methods adequately described?
Reply: The method section has been improved with some details (lines 150-159).
Are the conclusions supported by the results?
Reply: We thank the referee for this observation. In Paragraph 5 “Conclusions” we have modified as follow:
“Furthermore, digital PCR on cftDNA provides a promising and non-invasive assay to test EGFR mutations, also providing quantitative data. Compared to other methods digital PCR would be a robust method for absolute quantification of samples in a longitudinal way, allowing to better monitor the evolution of mutations over time. Moreover, our results highlight the potential application of liquid biopsy using Digital PCR as a routine assay in clinical practice for both detection and quantification of actionable mutation landscape in NSCLC patients.” (Lines 349-355)
Answers to specific concerns:
1) Therascreen and PANAMutyper are product names whereas Digital PCR is a name of methods. Therefore, to avoid reader’s confusion, names of methods but not products should be used throughout the manuscript. Alternatively, if the authors insist on product names, they should describe the reason in the introduction section.
Reply: We thank the referee for this observation. We have changed thought out the text the name of the methods.
2) Where did the numbers below come from?
Reply: we thank the referee for this observation. We have clarified as follow:
12/”12” in the line 96
“Specificity was calculated using DNA extracted from the plasma of 12 healthy blood donors. The results showed that specificity was 100% (12/12) for all the three mutations measured by each of the three methods (Fig. 1).” (Lines 193-195)
“104” in the line 139
In paragraph 3.5, we specified that the number 104 is the total number of EGFR mutations found in the cfDNA (line 243).
3) As to description in the line 133 – 137, results should be shown as Supplementary data.
Reply: We thank the referee for this observation. We have added the results of the limit of detection of the three methods for three mutations in the Supplementary Table S3. Corrected in the manuscript at lines 241.
4) As to Figure 2, it is difficult to interpret detection of T790M in plasma by digital PCR. How can we interpret detection of T790M in #1, #2, #5, #9, #23, and #24 Are they false positive? Dose the detection affect specificity of digital PCR?
Reply: Considering the high sensitivity of digital PCR we could hypothesize that T790M in those samples is underrepresented, or represented in a heterogeneous manner in tumor tissue, and it is detectable only with high sensitive methods such as Digital PCR.
5) As to Figure 3, it would be helpful for readers to show not only detected numbers of mutations but also examined total numbers in parentheses (e.g., 6/52, 1/22) even if examined total numbers are shown in the main text.
Reply: We agree with the reviewer. We have added in the figure 3 the number of mutations determined on the examined total number in parentheses.
6) The authors should define EV and PD in Figure 4 in more detail.
Reply: We have included in the figure legend these definitions (EV: first clinical evaluation, PD: disease progression).
7) Line 211 – 212: Only one T790M sample was tested in this study. Therefore, it would be difficult to conclude as “no difference” on detection of T790M.
Reply: We agree with the referee and we have added a specification (lines 218-219).
8) The discussion section is like a summary of results but not discussion. The authors should discuss results and pros/cons of digital PCR in more detail in the section.
Reply: A specific sentence concerning the potential usefulness of digital PCR in the clinical practice has been added in the Discussion (lines 334-342).
9) There is no detailed information on primer and probe sequences used for digital PCR. The information should be shown if the authors designed them by themselves. Alternatively, if the authors purchased primers and probes from Thermo Fisher Scientific as kits, those product numbers should be shown.
Reply: We agree with the reviewer. We have added in the section material and method, paragraph 2.7 the primer sequences and probes. We added the sentence: “We purchased all 3D Digital PCR reagents from Thermo Fisher Scientific,.” (lines 150-159).
10) PFS, HR, and OS should be defined in the line 186–187.
Reply: the acronyms have been defined (lines 169-171).
11) “4” and “*” are not shown in the line 6–8.
Reply: We have added the symbol “*” next to the name of Luigi Pasini who is the corresponding author and a symbol “#” next to the two authors who equally contributed to the work (line 6).

Reviewer 2 Report
Detection of EGFR mutations in plasma cell-free 2 tumor DNA of TKI-treated advanced-NSCLC patients 3 by three methodologies: Therascreen, PANAMutyper, 4 and Digital PCR. 5
Comments:
This paper does a nice job looking at 3 different methodologies of plasma cell-free DNA monitoring in NSCLC patients, Therascreen, PANAMutyper, and Digital PCR. I think many of the figures were well done and easy to follow.
My minor comments on this paper would be the following:
If possible, I would like to see the authors come up with a plausible explanation on why there were such differences in the sensitivity rates for the Del19.
I did not see a mention whether the tissue NGS was standardized or was multiple different testing platforms
Finally, I would prefer to see the discussion section at the end, with the conclusion, instead of sandwiching the materials and methods in between the discussion and the conclusion. I would like to see a statement that summarizes whether the technology appears to be appropriate for diagnosis and separately for MRD surveillance. I think it would also be important to emphasize whether this would apply to patients in the earlier stage disease (especially in the light of the ADAURA results), as a majority of their samples came from stage IV patients.

Author Response
Answers to reviewers
Manuscript: Siggillino et al - diagnostics-1001893
We thank the reviewer for the precious suggestions and comments provided. Please find our detailed answers in the text below. Correction and chances in the main manuscript are highlighted in yellow. We have also included a modified version of Fig. 1, Fig. 2, and Fig. 3, and added the Supplementary Table S3.
REVIEWER 2
Detection of EGFR mutations in plasma cell-free tumor DNA of TKI-treated advanced-NSCLC patients by three methodologies: Therascreen, PANAMutyper, and Digital PCR.
Does the introduction provide sufficient background and include all relevant references? can be improved
Reply: The introduction has been improved with new bibliographic references (number 17-18-19,25,26 – lines 416-423, 437-442).
Are the methods adequately described? Can be improved
Reply: We improved the method section with more details (lines150-159).
Comments:
This paper does a nice job looking at 3 different methodologies of plasma cell-free DNA monitoring in NSCLC patients, Therascreen, PANAMutyper, and Digital PCR. I think many of the figures were well done and easy to follow.
My minor comments on this paper would be the following:
If possible, I would like to see the authors come up with a plausible explanation on why there were such differences in the sensitivity rates for the Del19.
Reply: This an important point raised by the reviewer. We do not have a clear biological explanation to this discrepancy. Most probably, the different type of probes used in the three platform (for Therascreen and PANAMutyper we cannot provide the detail of probe design and the exact match on DNA) is the reason of this different sensitivity.
I did not see a mention whether the tissue NGS was standardized or was multiple different testing platforms.
Reply: On tumor tissue we did not use NGS to characterize patients’ mutations but two standardized methodologies routinely used in diagnostics, as mentioned in the methods (Scorpion-ARMS and MASSArray Sequenom) (lines 112-113).
Finally, I would prefer to see the discussion section at the end, with the conclusion, instead of sandwiching the materials and methods in between the discussion and the conclusion.
Reply: We have modified the manuscript structure accordingly.
I would like to see a statement that summarizes whether the technology appears to be appropriate for diagnosis and separately for MRD surveillance. I think it would also be important to emphasize whether this would apply to patients in the earlier stage disease (especially in the light of the ADAURA results), as a majority of their samples came from stage IV patients.
Reply: We thank the reviewer for this interesting comment. We have added a sentence in the Discussion underlying the possible implication of such a sensitive methodology in early-stage diseases (lines 342-345).